# Circulating Tumor Cell Subpopulations Predict Treatment Outcome in Pancreatic Ductal Adenocarcinoma (PDAC) Patients

**DOI:** 10.3390/cells12182266

**Published:** 2023-09-13

**Authors:** Ian M. Freed, Anup Kasi, Oluwadamilola Fateru, Mengjia Hu, Phasin Gonzalez, Nyla Weatherington, Harsh Pathak, Stephen Hyter, Weijing Sun, Raed Al-Rajabi, Joaquina Baranda, Mateusz L. Hupert, Prabhakar Chalise, Andrew K. Godwin, Malgorzata A. Witek, Steven A. Soper

**Affiliations:** 1Department of Chemistry, The University of Kansas, Lawrence, KS 66047, USA; ifreed@stowers.org (I.M.F.); damifat@ku.edu (O.F.); m367h922@kumc.edu (M.H.); pgonzalez@stetson.edu (P.G.); nweatherington@gmail.com (N.W.); mwitek@ku.edu (M.A.W.); 2Center of Bio-Modular Multiscale Systems for Precision Medicine (CBM^2^), The University of Kansas, Lawrence, KS 66047, USA; agodwin@kumc.edu; 3Division of Medical Oncology, University of Kansas Medical Center, Kansas City, KS 66160, USA; wsun2@kumc.edu (W.S.); ral-rajabi@kumc.edu (R.A.-R.); jbaranda@kumc.edu (J.B.); pchalise@kumc.edu (P.C.); 4Kansas Institute for Precision Medicine, University of Kansas Medical Center, Kansas City, KS 66160, USA; hpathak@kumc.edu (H.P.); shyter@kumc.edu (S.H.); 5Department of Cancer Biology, The University of Kansas Medical Center, Cancer Center, Kansas City, KS 66160, USA; 6Department of Pathology and Laboratory Medicine, University of Kansas Medical Center, Kansas City, KS 66160, USA; 7BioFluidica, Inc., San Diego, CA 92121, USA; matt@biofluidica.com; 8Bioengineering Program, The University of Kansas, Lawrence, KS 66045, USA; 9Department of Mechanical Engineering, The University of Kansas, Lawrence, KS 66045, USA

**Keywords:** circulating tumor cells, pancreatic cancer, PARP inhibitors, microfluidics, next-generation sequencing

## Abstract

There is a high clinical unmet need to improve outcomes for pancreatic ductal adenocarcinoma (PDAC) patients, either with the discovery of new therapies or biomarkers that can track response to treatment more efficiently than imaging. We report an innovative approach that will generate renewed interest in using circulating tumor cells (CTCs) to monitor treatment efficacy, which, in this case, used PDAC patients receiving an exploratory new therapy, poly ADP-ribose polymerase inhibitor (PARPi)—niraparib—as a case study. CTCs were enumerated from whole blood using a microfluidic approach that affinity captures epithelial and mesenchymal CTCs using anti-EpCAM and anti-FAPα monoclonal antibodies, respectively. These antibodies were poised on the surface of two separate microfluidic devices to discretely capture each subpopulation for interrogation. The isolated CTCs were enumerated using immunophenotyping to produce a numerical ratio consisting of the number of mesenchymal to epithelial CTCs (denoted “Φ”), which was used as an indicator of response to therapy, as determined using computed tomography (CT). A decreasing value of Φ during treatment was indicative of tumor response to the PARPi and was observed in 88% of the enrolled patients (n = 31). Changes in Φ during longitudinal testing were a better predictor of treatment response than the current standard CA19-9. We were able to differentiate between responders and non-responders using ΔΦ (*p* = 0.0093) with higher confidence than CA19-9 (*p* = 0.033). For CA19-9 non-producers, ΔΦ correctly predicted the outcome in 72% of the PDAC patients. Sequencing of the gDNA extracted from affinity-selected CTC subpopulations provided information that could be used for patient enrollment into the clinical trial based on their tumor mutational status in DNA repair genes.

## 1. Introduction

Pancreatic ductal adenocarcinoma (PDAC) is currently the third-leading cause of cancer deaths in the US [1]. The majority (~80%) of PDAC patients are diagnosed with advanced or inoperable disease and have a five-year survival rate of ~5% [2]. Attempts to improve the overall survival (OS) of PDAC patients with the current standard of care have been relatively unsuccessful [3]. Due in part to the lack of different precision therapies that match the molecular composition of a patient’s tumor, difficulty in securing solid tissue for testing actionable molecular targets, and a lack of robust biomarkers to track tumor response to therapy, PDAC is projected to become the second most common cause of cancer death in the U.S. [1]. Therefore, a high clinical unmet need exists to improve outcomes for PDAC patients either with the discovery of new therapies or accessible biomarkers to rapidly track response to treatment.

A major obstacle in monitoring treatment response in PDAC is the paucity of reliable predictive biomarkers or imaging methods that can be used for tumor evaluation. Following the start of systemic chemotherapy, oncologists wait 2–3 months for a radiographic result, typically based on tumor size, to determine disease response to therapy. Unfortunately, the determination of tumor size for PDAC patients using radiographic imaging can be affected by the dense pancreatic tumor stroma and the inability to detect <1 cm feature changes [4]. A blood-based biomarker recommended by the National Comprehensive Cancer Network and the FDA is carbohydrate antigen 19-9 (CA 19-9) [5]. Unfortunately, an elevated CA19-9 level (>37kU/L) of CA19-9 can be associated with other gastrointestinal malignancies, biliary obstruction, and pancreatitis [6,7]. Moreover, 6–20% of the general population does not produce CA19-9 [8]. CA19-9 was used as a marker of *recurrence* for PDAC patients: for those in which a 2-fold increase in CA19-9 serum level was detected, the CA19-9 test had a 90% PPV for recurrence, albeit with 45% clinical sensitivity and 85% specificity [6]. To improve the PPV, CA19-9 is typically used in conjunction with computed tomography (CT) to determine treatment responses [6]. Therefore, better biomarkers that can predict response to therapy sooner compared with CT and more reliably than CA19-9 are needed.

PDAC, as an epithelial cancer [9], releases tumor cells from primary and metastatic sites into blood (i.e., circulating tumor cells, CTCs). Analysis of CTCs presents an opportunity to assign treatment based on the molecular profile of these cells as well as track response to therapy. Our group demonstrated [10] 100% clinical sensitivity for CTC detection in the blood of PDAC and other epithelial tumors by selecting orthogonal subpopulations of CTCs that better recapitulate the complex PDAC tumor microenvironment. We targeted CTCs expressing either EpCAM (epithelial cell adhesion molecule) or FAPα (fibroblast activation protein alpha), which represent epithelial and mesenchymal cells, respectively. EpCAM is a well-recognized antigen for the affinity selection of CTCs. FAPα is a cell surface gelatinase that plays a role in facilitating cell invasion into the extracellular matrix and is expressed in >90% of human epithelial cancers, and cells expressing this antigen show a mesenchymal phenotype [11]. Selection of CTCs exclusively based on EpCAM can miss CTCs that undergo an epithelial to mesenchymal transition (EMT) [12]. The use of an orthogonal marker to EpCAM, which we present to be FAP, allows for better disease staging and monitoring of the tumor response to treatment even if EpCAM is downregulated. CTCs exist as a spectrum of phenotypes [13], which possess metastatic and therapy-resistant characteristics [14,15]. Several studies demonstrated CTCs as prognostic markers [10,16,17]. As CTCs can be obtained from a minimally invasive blood draw, it can lead to more frequent testing and thus predict non-responders to therapy quicker than CT. In contrast to other biomarkers, such as circulating tumor DNA (ctDNA) and extracellular vesicles (EVs), CTCs provide unique information regarding the correlation between phenotype and genotype.

The objectives of this study were: (i) evaluate whether changes in CTC burden across two orthogonal subpopulations can predict treatment response in patients diagnosed with metastatic PDAC (M-PDAC); (ii) compare the use of CTCs with CA19-9 in predicting treatment response; and (iii) determine if the mutational status of the tumor can be deduced using both CTC subpopulations. We used our CTC assay on M-PDAC patients undergoing treatment with a PARPi, niraparib (GSK3985771). PARPis are aromatic compounds that compete with NAD^+^ to inhibit the ability of PARP-1/2 to form polymer chains for DNA repair [18]. M-PDAC patients were enrolled in this trial if they harbored mutations in DNA repair genes and received oral 200 mg or 300 mg dose of niraparib (NIRA-PANC, https://clinicaltrials.gov/ct2/show/NCT03553004 accessed on 15 December 2022).

CTCs were isolated directly from whole blood using a microfluidic based on affinity selection in an automated fashion using a liquid-handling robot fitted to distribute fluids to the microfluidic chips. We evaluated the mutational status of the CTCs from their gDNA at baseline and at the end of treatment (EOT) using next-generation sequencing (NGS) and ligase detection reactions (LDRs) for the detection of DDR (DNA damage response) and *KRAS* mutations, respectively. This study longitudinally tracked CTCs in 31 patients (median enrollment time of 107 days, blood sampling every ~28 days).

## 2. Materials and Methods

### 2.1. Study Design

Metastatic PDAC patients enrolled in this study were pre-screened for tumor tissue mutational status in the clinical molecular oncology lab at the University of Kansas Medical Center (KUMC) to confirm the presence of germline or somatic mutations in DNA-repair genes (Appendix A). The enrolled patients received previous first- and/or second-line therapy. Baseline CT scans were taken prior to starting PARPi therapy and approximately every 8 weeks thereafter. Blood for CTC analysis was drawn every 4 weeks into EDTA purple top tubes, at the beginning of each cycle; blood CA19-9 was also measured at this time. CTCs were isolated on microfluidics from whole blood for both enumeration and gDNA extraction.

### 2.2. Blood Sample Processing

Patient blood samples were provided by the University of Kansas Medical Center (KUMC) under approved IRB. Blood (~7 mL) was collected into a spray-coated K_2_EDTA tube and was additionally stabilized with an eptifibatide (Sigma-Aldrich, St. Louis, MO, USA) [19] to a final concentration of 50 μg/mL to allow for up to 72 h of cold blood storage post-collection. Samples were processed on antibody-modified microfluidic chips mounted on a Hamilton Microlab Starlet Robotic system (Hamilton Company, Reno, NV, USA) (Figure 1A,B). Functionalized chips were first washed with a blocking buffer (0.5% BSA in PBS pH 7.4) (BSA: Sigma-Aldrich; PBS: Gibco, Grand Island, NY, USA). Then, up to 2 mL of whole patient blood (some samples did not have enough blood for 2 mL processing) was processed through each chip at a linear velocity of ~2 mm/s followed by a wash with 0.5% BSA in PBS solution at 4 mm/s. Processing larger volumes of blood is possible using our technology, which would result in a higher number of isolated CTCs, albeit at the cost of longer analysis time. As we collected sufficient numbers of CTCs to secure enumeration data and molecular analysis from 2 mL blood, we believe that further increasing the blood volume (above 7 mL) from patients undergoing treatment would present challenges. Total processing time per sample was ~1.5 h. To enumerate and visualize captured CTCs, the captured cells are fixed and stained for target markers. Prior to fixation, the surfaces of cells were stained by filling the device with anti-CD45-FITC Ab staining solution (R&D Systems, Minneapolis, MN, USA) (2.5 μg/mL) and allowing it to incubate for 40 min at 4 °C. The chip was washed with PBS pH 7.4. Cells were then fixed for 15 min on a chip by infusing 4% formaldehyde in PBS pH 7.4. The chips were washed again with PBS for 5 min. All washings were performed at 50 µL/min and then infused with a solution of 0.1% Triton-X100 (Sigma-Aldrich) in water and 20 μg/mL 4′,6-diamidino-2-phenylindole (DAPI) and incubated for 10 min at RT to porate the cell membrane and provide staining of the nucleus. The chips were then washed with PBS for 2 min. Then, the captured cells were stained using a mixture of pan-cytokeratin (pan-CK) (R&D Systems) (6 μg/mL) and vimentin (VIM) (R&D Systems) (4 μg/mL) for 40 min at 4 °C. After incubation, the chips were again rinsed with PBS for 5 min. Following on-chip immunophenotyping, the chips were imaged using a Zeiss 200M Axiovert (Carl Zeiss, White Plains, NY, USA) microscope equipped with an XBO-75 Xenon lamp, and images were taken with a Cascade 1K CCD camera (Roper Scientific, Tucson, AZ, USA). The images were processed using ImageJ software (National Institute of Health, Bethesda, MD, USA).

### 2.3. Polymerase Chain Reaction and Ligase Detection Reaction (PCR/LDR) Assay

PCR was performed with WGA DNA in a total volume of 50 μL using OneTaq 2X Master Mix with Standard Buffer (New England Biolabs, Ipswich, MA, USA). PCR cocktails consisted of 5 μL of primers, 25 μL OneTaq 2X Master Mix with Standard Buffer, 16 μL nuclease-free water, and 4 μL gDNA. The PCR was carried out in a thermal cycler (MJ Research Inc., Waltham, MA, USA) with the following steps: denaturation at 94 °C for 30 s followed by 30 cycles of denaturation at 94 °C for 15 s; annealing for 30 s at 58 °C and extension at 68 °C for 1 min. A final extension at 68 °C for 5 min was followed by a cooling step at 4 °C. *KRAS* primers were obtained from IDTDNA: forward primer 5′ *AAC CTT ATG TGT GAC ATG TTC TAA TAT AGT CAC* 3′ and reverse primer 5′ *AAA ATG GTC AGA GAA ACC TTT ATC TGT ATC*- 3′. The presence of 290 bp long amplicons was confirmed and quantified using an Agilent 2200 TapeStation. The LDR was performed in 20 μL reaction volume with HiFi Taq DNA Ligase (New England Biolabs, Ipswich, MA, USA). The LDR cocktail contained discriminating and common primers 4 nM each, amplicons 0.6–1 ng (3–5 fmol), 40 units of DNA ligase, and buffer containing cofactor. Thermocycling conditions were: 95 °C for 1 min and 63 °C for 2 min for 20 cycles. Common primers for *KRAS* codons 34 and 35 were 3′ Cy5-labeled. Discriminating primers were designed to produce ligated products of different sizes. LDR products were separated using a Beckman CQ CE system and sized against the ladder (608395, Beckman Coulter, Brea, CA, USA). The injection was performed at 2kV for 1 min, and separation was performed at 7.5 kV for 30 min run time. The capillary ID was 75 µm, and the length was 33 cm (Beckman Coulter).

Control PCR/LDR experiments were set up using gDNA isolated from cell lines with known *KRAS* mutations (HT29, *KRAS* wild type; RPMI, KRAS c.35G>C, p.G12A; SW480, *KRAS* c.35G>T, p.G12V; HCT116, *KRAS* c.G38G>A, p.G13D). The cell lines were a generous gift from Prof. Dan Dixon at KU.

### 2.4. Next-Generation Sequencing (NGS)

For formalin-fixed paraffin-embedded tumor tissue samples, gDNA was isolated using QIAamp DSP DNA FFPE Tissue Kit (Qiagen, Germantown, MD, USA). For buffy coat samples, gDNA was isolated using the DNeasy Blood & Tissue Kit (Qiagen), and cell-free DNA was isolated from 1 mL plasma samples using the QIAamp Circulating Nucleic Acid Kit (Qiagen). DNA was quantified using a NanoQuant Plate on an Infinite M200 Pro plate reader (Tecan, Männedorf, Switzerland) and then underwent library preparation using a multiplex PCR reaction targeting exons within 33 DNA repair-related genes (custom QIAseq Targeted DNA panel, Qiagen). The quality of the prepared libraries was assessed using DNA1000 TapeStation Analysis (Agilent, Santa Clara, CA, USA), and the quantity was assessed using the QIAseq Library Quant Assay Kit for Illumina libraries (Qiagen). The prepared libraries were then subjected to next-generation sequencing on a NextSeq 550 instrument (Illumina, San Diego, CA, USA) to generate FASTQ files. The reads were mapped to GRCh37 reference using the CLC Genomics Workbench (Qiagen) to generate variant call files, which were processed using Clinical Insight-Interpret Translational (Qiagen) to assess pathogenicity based on ACMG-AMP guidelines [20]. Quality control metrics such as depth of coverage, variant allele frequency, and average quality scores for the reported variants were evaluated individually for pathogenic and likely pathogenic calls compared to variants of uncertain significance.

### 2.5. Statistical Tests

Wilcoxon-signed rank tests were applied given the non-normal distribution of paired samples for comparing different subpopulations of CTCs from the same patient(s). Mann–Whitney U-tests were used to compare the number of non-CTCs in HD and patients. Statistical testing was performed using SPSS software (IBM Corporation, Armonk, NY, USA).

## 3. Results

### 3.1. Automated Blood Sample Processing Using the Microfluidic CTC Isolation Device

The microfluidic chips used in this study were injection molded from cyclic olefin polymer (COP). The microfluidic contained 150 sinusoidal microchannels with a z-architecture equipped with an inlet and outlet port designed to receive pipette tips (Figure 1A). Device surfaces were covalently decorated with a single mAb type; one assay consisted of processing a blood sample on both anti-EpCAM (R&D Systems, clone#158210) and anti-FAPα (R&D Systems, clone#427819) modified chips. Blood entered the CTC selection device through a single inlet channel, passed through a parallel array of sinusoidal mAb-laden selection channels, and exited through a single outlet channel. After blood processing and washing, each CTC subpopulation on separate chips was interrogated independently; these data would have been obscured by immobilizing both mAbs in one device. The detailed design of the chip was discussed previously [21]. An automated liquid-handling robot with custom software and a work deck was used to operate up to eight microfluidic chips simultaneously (Figure 1B). The blood was processed directly without pre-processing such as the removal of red blood cells. The robot delivered liquid to the chips using an air displacement mode via pipettes that were inserted into the inlet and outlet ports of the chip. Fluid delivery to the microfluidic network was achieved by simultaneously operating two pipetting channels, one in “push” and another in “pull” mode. The system is flexible in terms of volumes that can be delivered and liquid flow rates, which are programmed into the operating protocol using the graphical user interface (Figure 1C). Simultaneous use of two pipetting channels led to uniform sample and reagent flow.
Figure 1Microfluidic device and robotic system. (**A**) Photograph showing a cell isolation chip with custom inlet and outlet ports designed to receive pipette tips. SEM images showing chip architecture. The microfluidic chip contains 150 channels 2 cm long (25 μm × 150 μm, w × d) with a sinusoidal design to exploit centrifugal forces that drive cells toward the channel walls, increasing the capture rate when compared with straight channels. The device has tapered inlet and outlet channels to allow a constant linear velocity in each channel and control shear stress. The footprint of the chip is 43 × 40 mm (w × l). The chip was injection molded in cyclic olefin polymer (COP). (**B**) Sample handling is automated using a Hamilton Starlet Robotic system with custom software. The pipetting arm is equipped with 1 mL or 0.3 mL pipettes (1) that move in three dimensions (x, y, z) to aspirate and dispense liquid samples. Samples are loaded from test tubes at (2) and delivered to chips that are placed at (3). Samples are delivered to chips using air displacement in which one pipette (1 and 3) dispenses the sample while the other (2 and 4) aspirates through the chip. (**C**) Software parameters established for processing blood samples. (**D**) Performance metrics of the robot in dispensing fluids.
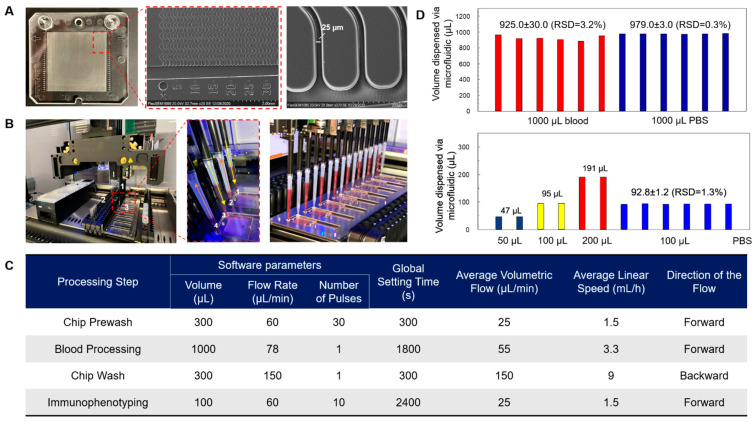


The entire system was fully automated, providing ease of use and reduced operator variation and offering excellent reproducibility and accuracy. The blood sample processing rate used herein was 3.3 mL/h and a post-wash of 9 mL/h. For on-chip staining, the liquid-handling robot could dispense immunophenotyping reagents, and with incubation, it required ~1.5 h processing time for a set of surface and cytosolic markers.

Previous work [10] by our group used similar microchip devices but with 50 microfluidic channels and a syringe pump for sample introduction, requiring manual handling of both reagents and blood samples. The main advantage of blood processing using the robotic system was its automated format and ability to infuse all the blood samples through the CTC isolation chip, as the pipette-to-chip interface provides a near-zero dead volume connection. Owing to a dead volume of ~100 µL using syringe-to-chip connectors, processing samples <100 µL was difficult. However, using the robot, <100 µL blood or plasma samples were easily processed. The accuracy for dispensing blood volume was found to be 89–92%. Figure 1D and Appendix A summarize the performance metrics of the robot. Cell recovery using the robot was 68 ± 27% for the EpCAM-bearing cell line SKBR3 with an anti-EpCAM antibody-modified chip and was not statistically different when processing the same sample using a syringe pump. The recoveries were as we reported previously for this chip design and SKBR3 cells [10,21]. Two microfluidic chips were used for each assay, one chip containing anti-EpCAM antibodies and a second with anti-FAPα antibodies. The number of CTCs in each chip was used to determine the CTC burden for each subpopulation, CTC^EpCAM^ and CTC^FAPα^, respectively.

### 3.2. Isolation of CTCs from Blood Samples

M-PDAC patient enrollment into the PARPi (niraparib) trial was predicated on germline or somatic mutations in DDR genes present in tumor gDNA (Figure 2A). When the PDAC tumor tissue harbored at least one mutation in the DDR panel (Appendix A), a patient was eligible to receive niraparib and continued treatment for up to nine cycles if CT images indicated continued reductions in tumor size.

Blood samples were analyzed for the presence of two CTC subpopulations. CTCs from each subpopulation were affinity-selected based on antigens present within their cellular membrane, i.e., those expressing EpCAM or FAPα, namely, CTC^EpCAM^ or CTC^FAPα^ (Figure 2B), respectively [10]. A blood sample (2 mL per chip) was processed on two chips, each decorated with a mAb targeting epithelial-like or mesenchymal-like CTCs. Following isolation, CTC counts for both subpopulations were tallied. Upon initial inspection of the number of CTC^EpCAM^ and CTC^FAPα^ as a function of the treatment cycle number of our patient cohort, no clear relationship between CTC number and response to therapy was evident for either subpopulation alone, as identified using CT (see Appendix A). Therefore, we sought to use the ratio of CTC^FAPα^ to CTC^EpCAM^, Φ (Figure 2C), as a metric for tumor response to therapy.

CTCs were identified using immunophenotyping (Appendix A). For a cell to be considered a CTC, it needed to demonstrate the presence of a nucleus with a positive DAPI signal, expression of pan-cytokeratins (CK) (++/+/−), or vimentin (VIM) (+/−), and lack of leukocyte-specific antigen CD45 (−) (Figure 2D,E). We also enumerated released CTCs based on impedance sensing (Appendix A), a method that was previously reported and validated by our group [10]. The advantage of impedance sensing is that the enumerated and non-stained cells can be further processed for gDNA analysis, while cells that are fixed and stained cannot be used for molecular profiling, owing to potential changes in DNA and/or RNA integrity induced by crosslinking agents [22].

The highest burden of both CTC^EpCAM^ and CTC^FAPα^ subpopulations was detected at baseline (before treatment administration on day 1 of cycle 1, c1d1). The averages were 27/mL and 21/mL for CTC^EpCAM^ and CTC^FAPα^, respectively (n = 27). In many cases, the enumeration of CTCs during subsequent cycles indicated a decrease in the number of both CTC^EpCAM^ and CTC^FAPα^. The positivity of the test was recorded when counts were above the threshold values determined based on analysis of healthy donor (HD) blood samples (Figure 2F, Appendix A).

As not all patients completed the same number of treatment cycles based on the results from their CT scans, the number of patients in the second column in Figure 2F varies. “EOT” is end of treatment and is the last CTC test performed. It occurred when a patient discontinued trial participation due to disease progression, withdrawal from the trial, or death.

HD blood (n = 11) was used as a control to determine assay specificity. An average of 0.3 ± 0.7 cells/mL showed staining for pan-CK on the anti-FAPα modified microfluidic chip and 0.4 ± 0.9/mL cells on the anti-EpCAM mAb modified chip from HD blood samples (Appendix A). Clinical sensitivity was calculated at each treatment cycle and was established when the number of CTCs enumerated exceeded the thresholds observed in the HD samples (Appendix A). Less than 1 white blood cell (WBC)/mL identified as DAPI (+)/CK (−)/VIM (−)/CD45 (+) was isolated on both the anti-FAPα and anti-EpCAM mAb chips (n = 11; Appendix A). The number of cells identified as DAPI (+)/CK (−) /VIM (+)/CD45 (−) that were isolated on the anti-FAPα and anti-EpCAM chips in the HD samples was 3.3 and 4.0 cells/mL, respectively; in the patient samples, the medians for this cell type were 2.5 and 3.0 cells/mL on the anti-FAPα and anti-EpCAM chips, respectively (n = 68). There was no statistical difference found between the total number of this CTC type isolated when comparing patient samples and HD samples (*p* = 0.849, anti-FAPα; *p* = 0.313, anti-EpCAM, Mann–Whitney U-tests). This type of CTC was indicative of either fibroblasts, macrophages, or adipocytes (non-CTC type) and possibly a mesenchymal CTC type with CK expression that was downregulated [23,24]. CTCs that were DAPI (+)/CK (+)/VIM (−)/CD45 (−) were delineated as epithelial CTCs, DAPI (+)/CK (+)/VIM (+)/CD45 (−) phenotypes were classified as epithelial/mesenchymal hybrid CTCs, and those that were DAPI (+)/CK (−)/VIM (+)/CD45 (−) were labeled as non-CTCs.

In the patient samples, the purities for selected CTC subpopulations were 61.4% (n = 40) and 61.8% (n = 37) for CTC^FAPα^ and CTC^EpCAM^, respectively, when processed using the robot (Appendix A), and were calculated as the number of CTCs divided by the total number of cells enumerated on a chip (CTC + non-CTC + WBC). Upon CTC contact with the antibody-decorated microfluidic wall, CTCs are affinity-selected. Specific antigen-antibody interactions can withstand high blood shear forces generated in the microchannels, while WBCs under these conditions cannot bind effectively to the wall, abating the number of non-specific adsorption artifacts and providing high purity of isolated CTCs. Total CTC counts along with purity are provided in Appendix A**.**

Clinical sensitivities for CTCs were determined for both subpopulations at each treatment cycle. The lowest CTC sensitivities were detected in cycle 3 of the PARPi treatment: 87.5%, 68.8%, and 87.5% for CTC^FAPα^, CTC^EpCAM^, and combined CTC^FAPα^ and CTC^EpCAM^, respectively. The average sensitivities for all cycles with combined CTC subpopulations was 97.6% (Appendix A). Based on clinical sensitivities and specificities, we determined the ROC for CTC^FAPα^ and CTC^EpCAM^ separately and in combination (see Appendix A), which suggested that test positivity increased when both CTC subpopulations were enumerated. The ROC analysis is presented in Appendix A. This analysis showed an AUC of 94.2%, 94.6%, and 96.5% for CTC^FAPα^, CTC^EpCAM^, and both CTC^FAPα^ and CTC^EpCAM^, respectively.

Our staining panel included both pan-CK and VIM to account for phenotypic differences between CTC^EpCAM^ and CTC^FAPα^. Cells expressing VIM were more prevalent in the CTC^FAPα^ subpopulation, with 90% of CTC^FAPα^ showing a detectable signal for VIM in addition to pan-CK. Overall, 69% of CTC^EpCAM^ showed the presence of both pan-CK and VIM (hybrid CTCs), and ~30% of CTC^EpCAM^ showed pan-CK exclusively (*p* = 0.000033, Wilcoxon-signed rank, n = 57 samples, Appendix A). When CTC^EpCAM^ and CTC^FAPα^ staining patterns were compared at different treatment cycles, a notable difference in the expression of pan-CK and VIM was observed during baseline testing on c1d1, where 95% of CTC^FAPα^ expressed VIM compared to 63% of CTC^EpCAM^ (*p* = 0.046, Wilcoxon-signed rank, n = 8; Figure 2G). For CTC^EpCAM^ between cycles 1 and 2, we found a significant difference in the percent of CTC^EpCAM^ with VIM expression: 63% in cycle 1 and 81% in cycle 2 (*p* = 0.043, n = 7 for paired samples). Furthermore, there was no significant difference in the percent fraction of CTC^EpCAM^ with VIM expression during further treatment.

### 3.3. CTC^FAPα^ to CTC^EpCAM^ Ratio (Φ) and the Treatment Response

The CTC^FAPα^ to CTC^EpCAM^ ratio (Φ) was analyzed as a measure of tumor response or lack thereof to niraparib treatment. The changes in Φ (ΔΦ) were assessed during each treatment cycle. ΔΦ was calculated as N_+1_ (Φ) − N (Φ), where N is the treatment cycle number and N_+1_ is the subsequent cycle number and could be assessed if at least two longitudinal CTC measurements were performed. The ΔΦ value was considered significant when Φ between cycles changed by ≥ 20%. ΔΦ was compared to results for both CA19-9 and CT imaging at treatment cycles. It was noted that when CT imaging showed the progression of a PDAC tumor, the value of Φ increased (ΔΦ was positive). When tumor size decreased, as observed from the CT image, the value of Φ decreased between treatment cycles or no change above 20% was observed (ΔΦ was a negative value or <20%). Based on these results, we were able to differentiate between stable (responders) and progressive (non-responders) disease using ΔΦ (*p* = 0.0093, n = 47) with higher confidence than using CA19-9 (*p* = 0.033, n = 40) (Figure 3A,B). To construct these plots, we used patient data for which a CT scan and CA19-9 tests were performed at the same time as CTC enumeration.

Evaluation of ΔΦ, CA19-9, and CT imaging (Appendix A) showed a correlation between ΔΦ and CT scans for 22 (88%, Appendix A) patients for whom both ΔΦ and CT imaging was performed (n = 25). For three patients, there was no correlation; ΔΦ was negative, but CT imaging showed progressing disease (Appendix A). All patients shown in Appendix A had a positive ΔΦ value that preceded or coincided with disease progression, as determined using CT.

We evaluated whether ΔΦ and ΔCA19-9 correlated in predicting tumor progression, as observed using CT **(**Appendix A). Out of 25 patients with multiple CA19-9, Φ, and CT testing, 7 patients did not produce CA19-9 or had very high CA19-9 levels with immeasurable changes during treatment (Appendix A). In five out of seven (72%) patients with immeasurable levels of CA19-9, ΔΦ was informative and correlated with CT, showing either stable disease or progression. Overall, in 10/31 PDAC patients, CA19-9 had 0% PPV (Appendix A). ΔΦ and ΔCA19-9 (N_+1_ (CA19-9) − N (CA19-9)) correlated in 40% of the patients (Appendix A), with no consensus in 36% (Appendix A).

For six patients in this study, we enumerated CTC subpopulations only once during treatment, so no ΔΦ could be derived. Those patients were not included in the ΔΦ analysis. However, while a single measurement disallowed the determination of ΔΦ, it did not prohibit the evaluation of disease status. A Φ > 1 for a single test implied a higher fraction of CTC^FAPα^ with reference to CTC^EpCAM^. In 20/30 patients tested at EOT for whom CT indicated progressing disease, Φ ranged between 1.2 and 39.0 (average of 5.8, median of 2.0, Appendix A).

Plots of CTC^FAPα^ and CTC^EpCAM^ counts, Φ, CA19-9, and CT results are shown for patients #6 (Figure 3C,D) and #11 (Figure 3E,F). The yellow and purple bars in the plots are indicative of stable and progressing disease, respectively, as determined using CT. For patient #6, stable disease was shown by both Φ and CT data from day 61 to day 180. Between days 180 and 208, Φ increased more than 11-fold (0.48 to 5.5). On day 243, disease progression was detected using CT (Figure 3G). CA19-9 was shown to increase throughout treatment. For patient #11, the tumor was non-responsive to treatment, as indicated by an increase in primary tumor size after 30 days of treatment (Figure 3H). Φ and CA19-9 increased significantly between days 0 and 30 of treatment.

### 3.4. Progression-Free Survival (PFS) and Overall Survival (OS)

The ability of ΔΦ and ΔCA19-9 to detect responders prior to CT was evaluated. Figure 4 shows a timeline of PDAC tumor progression observed for ΔΦ *vs* CT (Figure 4A) and ΔCA19-9 vs. CT (Figure 4B). For the data shown in Figure 4F, on average, patients that had measurable levels of CA19-9, ΔΦ and ΔCA19-9 predicted PDAC progression earlier than CT, after 29 ± 23 days and 21 ± 23 days, respectively.

When Kaplan–Meier plots were constructed for these two indicators, clear differences were observed. We evaluated time until EOT as an endpoint (i.e., PFS) and OS using both ΔΦ and CA19-9 markers. Kaplan–Meier plots were constructed for two cohorts: (*i*) patients for whom ΔΦ or ΔCA19-9 levels were >0 during the entire longitudinal testing period and (*ii*) patients for whom ΔΦ and ΔCA19-9 levels were <0 at least once during testing.

Based on the Kaplan–Meier plots, patients with ΔΦ > 0 during treatment had a lower OS compared with those who had ΔΦ < 0 at least once (*p* = 4.95 × 10^−4^; Figure 4C). The same was not true for ΔCA19-9. Patients with ΔCA19-9 > 0 through treatment and those with ΔCA-19.9 > 0 at least once had similar OS (*p* = 0.256; Figure 4D). Time to EOT (i.e., PFS) was also greater in patients with ΔΦ < 0 at least once during treatment (*p* = 1.49 × 10^−4^; Figure 4E). Decreases in CA19-9 levels have been associated with a more favorable patient survival [25]; however, in our study, ΔCA19-9 < 0 was not predictive of PFS (*p* = 0.434, Figure 4F). We further divided patients into three groups (Appendix A): (i) ΔΦ or ΔCA19-9 values were never <0 during treatment; (ii) ΔΦ or ΔCA19-9 values were <0 at least once; and (iii) ΔΦ or ΔCA19-9 values were <0 more than once during treatment. For these three groups, Kaplan–Meier OS and PFS were evaluated. When ΔΦ decreased more than once during treatment cycles, patients experienced longer PFS (*p* = 2.37 × 10^−5^; Appendix A).

### 3.5. Next-Generation Sequencing (NGS) of CTCs to Search for DDR Mutations

Genomic DNA (gDNA) extracted from CTC subpopulations was subjected to whole genome amplification, and the amplified gDNA was analyzed using NGS. A single-cell analysis was not performed. The high purity of isolated CTC subpopulations allowed for NGS from “bulk” CTCs. Appendix A shows representative electrophoretic traces of WGA amplicons for two patients: Pt #6 and Pt #2. The size of WGA gDNA ranged between 400 bp and 48 kbp for both CTC subpopulations. The size mode of WGA amplicons was ~10 kbp, indicating high-quality material. WGA yielded 100–500 ng of DNA compared with ~0.1 ng or less before WGA. Appendix A summarizes results from WGA of CTC gDNA for all patients.

Exon-targeted sequencing was used to test tumor DNA to allow for patients with DDR mutations to be enrolled in the clinical trial (Appendix A). In all cases, DDR mutations detected in tumor tissue were also found in the gDNA of CTCs for both subpopulations when tested at baseline, which suggested that we could use these liquid biopsy markers as enrollment criteria of patients into this clinical trial as opposed to a solid tissue biopsy.

In two patient samples submitted for NGS analysis (Figure 5), new mutations in DDR genes were detected at the EOT compared with the mutational status at baseline. Most notably, gDNA from both CTC^EpCAM^ and CTC^FAPα^ harbored mutations in *BRCA2* with ~36% variant frequency for patient #2 (ins A>pos.32907420^32907421) and ~46% variant frequency for patient #6 (SNV C>G, pos.32937504). The same mutation was detected in these patients’ buffy coats. At the EOT for patient #2, mutations in *BRCA2* were no longer detected in either CTC^EpCAM^ or CTC^FAPα^. In patient #6, the mutation in *BRCA2* was no longer detected in CTC^EpCAM^ but was present at high frequency in CTC^FAPα^ (Figure 4A). Also, at the EOT for both CTC subpopulations, new mutations were detected in *BARD1* (SNV C>G, pos.215617245) with > 98% frequency and high coverage (>12,000×; Figure 4B, Appendix A) for these two patients. However, *BARD1* SNV (c.1603G>C/p.D535H) is defined as a variant of uncertain significance (Clinvar accession VCV001319415.2).

The criteria for reporting mutations in CTCs included at least 250× coverage and >5% variant frequency or at least 300× coverage and >4% variant frequency. These criteria set the probability of a false negative at <0.5% [26]. In total, for patient #2 we saw 10 changes in variant frequency for CTC^EpCAM^ including *BRCA2*. In CTC^FAPα^ for patient #2, nine changes were seen. In CD4^+^ cells, a total of 13 changes were detected (Appendix A). In patient #6, we saw a total of 14 changes in CTC^EpCAM^, 23 for CTC^FAPα^, and 28 for CD4^+^ cells (Appendix A). There were seven “new” mutations for CTC^EpCAM^, eight for CTC^FAPα^, and 18 for CD4^+^ cells. Appendix A list all mutations documented in CTCs. We also performed NGS on patient buffy coats and cfDNA for comparison to data secured from CTCs. Germline mutations found in patient buffy coats were observed in both cfDNA and CTCs. The full data can be found in Appendix A.

### 3.6. KRAS Mutation Screening in CTCs Isolated from the Blood of PDAC Patients

gDNA isolated from CTCs was also tested for *KRAS* mutations. Following CTC isolation, gDNA was whole-genome amplified (WGA). NGS was performed with an Illumina Focus Panel on gDNA isolated from the SKBR3 cell line with and without WGA to verify no observed amplification bias (plotted coverage for WGA gDNA vs. non-WGA gDNA showed a slope of ~1; Figure 6A).

Screening for *KRAS* mutations was performed with a PCR/ligase detection reaction (PCR/LDR) assay (Figure 6B). An advantage of PCR/LDR is the fact that it can detect one SNV in 4000 wild-type templates with a high signal-to-noise ratio [27]. gDNA isolated from cell lines containing *KRAS* mutations (wild-types, c.35G>T, c.35G>A, and c.35G>C) was used as a positive control (Appendix A). Primers used for the LDR are shown in Appendix A. Figure 6C shows representative electropherograms collected using capillary gel electrophoresis for wild-type (c.35G) and mt (c.35G>T) ligated products following PCR/LDR. Green lines in the electropherograms represent 20 nt and 80 nt markers for LDR product sizing; the LDR products are represented by black line electropherograms. *KRAS* wild-type and products indicating mutations in gDNA extracted from CTCs are marked in the electropherograms. A high-intensity peak between 11.5 and 12 min represents unligated dye-labeled common primers used for the LDR. The bottom panels in Figure 6C show the detection of wild-type c.35G from CTC^EpCAM^.

CTCs isolated from nine patients were tested using PCR/LDR/CGE. *KRAS* wild-type (positive control) was confirmed in all patient samples tested (Figure 6D). Overall, 57% of the tested CTC samples had c.35G>T (p.G12V) and c.35G>C (p.G12A) mutations in PDAC gDNA. c.35G>A (p.G13D) or c.34G>C (p.G12R) mutations were not detected in any sample. Among nine patients tested, CTC gDNA from two patients (#5 and #33) had no detectable *KRAS* mutations. All other patients’ CTCs displayed *KRAS* mutations either at baseline or EOT. In patient #2, a c.35G>C (p.G12A) *KRAS* mutation was detected in the CTC^FAPα^ subpopulation at treatment onset; however, this mutation was not present in CTC^EpCAM^. At EOT, c.35G>C *KRAS* was still detected in CTC^FAPα^, but a new c.35G>T (p.G12V) *KRAS* was now detected in both CTC^FAPα^ and CTC^EpCAM^ subpopulations, possibly indicating the evolution of a different clone of tumor cells ~120 days after treatment.

## 4. Discussion

There are currently 17 studies in the U.S. that use CTCs as a biomarker for clinical trials associated with PDAC [28]. While CTCs offer unique attributes, there are other liquid biopsy markers that can be used in monitoring disease response to treatment. The challenge with ctDNA [29] is to effectively detect somatic mutations from the non-diseased cfDNA pool. EVs are generating interest [30], but analysis of their cargo is difficult, owing to the low mass of nucleic acids found within them and the few full-length mRNA transcripts they carry [31]. In contrast, CTCs following enrichment contain full-length genetic/transcriptomic signatures of the tumor, making molecular assays more reliable as long as the number of CTCs is sufficient and of high purity [10].

CTCs have been examined for their ability to track treatment response in clinical settings in a variety of cancers [32] and have largely used cut-off values to separate patients into ‘favorable’ and ‘unfavorable’ response groups. It has been argued that because both patient physiology and disease heterogeneity can play a large role in CTC burden, absolute cut-off values may be problematic [33]. Smerage et al. used the burden of epithelial-like CTCs in M-breast cancer patients to assign “high-risk” patients based on ≥5 CTC/7.5 mL blood to either maintain treatment or change to an alternative therapy. Their study found no difference in PFS or OS between patient cohorts [34]; however; they neglected the role of EMT in disease progression by enumerating only epithelial-like CTCs. Given the more recent literature on EMT, a broad range of CTC types should be considered in the analysis of response to therapy [12,13,14,15].

Unique to our assay is that mesenchymal and epithelial subpopulations of CTCs provide a disease response value Φ, which is based on two orthogonal CTC phenotypes, FAPα and EpCAM, expressing CTCs [10]. We evaluated the expression of mRNA transcripts associated with EpCAM and FAPα in isolated CTCs. The abundance of EpCAM mRNA transcripts was detected in CTC^EpCAM^ cells with FAPα mRNA undetected, and vice versa (undetectable EpCAM mRNA for CTC^FAPα^ and detectable FAPα mRNA only in the CTC^FAPα^ subpopulation). FAPα is a cell surface protease that plays a role in facilitating cell invasion into the extracellular matrix and is expressed in > 90% of human epithelial cancers. Cells that express FAPα have high tumor-initiating capabilities as opposed to epithelial-like cells. This protein is differentially expressed within cell membrane protrusions and can degrade a variety of substrates. From our previous work, > 90% of affinity-selected CTCs using EpCAM and FAPα did not co-express both antigens [10].

We designated CTC^EpCAM^ as cells captured within a microchip decorated with anti-EpCAM antibodies and CTC^FAPα^ in a similar chip but decorated with anti-FAPα antibodies, which were used to designate epithelial and mesenchymal CTCs, respectively, instead of using immunophenotyping of cellular markers (pan-CK and VIM) to make this designation. Each chip had some unique characteristics, including a large number of channels and large channel depth to increase throughput and large channel shear force to improve the purity of the chips, thus negating the need for single-cell picking for downstream molecular analysis [10]. Indeed, the chips coupled with FAPα and EpCAM affinity selection provided high purity, as only a few hematopoietic cells were co-isolated (Table S3). The sample processing workflow involved whole blood infusion into the CTC chips without the need for blood pre-processing. In addition, the CTC selection chips were made from a thermoplastic to allow for high-scale production using injection molding and were integrated into a liquid-handling robot to fully automate the process of samples.

The standard-of-care treatment response measurement in PDAC is CT imaging or variants thereof. For CT, subtle tumor mass changes can be lost due to similar tumor tissue and normal pancreas attenuation, small or scattered size of the tumor, and hidden location in the uncinate process [35]. Moreover, in locally advanced PDAC, it is hard to assess a radiographic response due to a desmoplastic reaction that can mask the true tumor volume. Finally, it is difficult to differentiate necrosis, fibrosis, or edema from residual tumors. Hence, alternative methods for assessing treatment response are of high clinical need.

While the blood-based biomarker CA 19-9 is recommended as an indicator for PDAC patient treatment response [5], there is no consensus on its utility. It was found that for 12% of patients who have persistent CA19-9 levels < 10 kU/l, monitoring CA19-9 offers 0% PPV [6]. A retrospective analysis found that a ≥20% decrease in CA19-9 burden was predictive of greater PFS [36], but other studies have shown that serum levels of CA19-9 in patients receiving gemcitabine never decreased despite patients being in complete remission [37]. In another trial including PDAC patients undergoing two different treatment regimens, no significant differences in PFS predictability were reported based on CA19-9 levels. While all patients with tomographically confirmed tumor regression had significant CA19-9 decrease, another group of patients with decreasing CA19-9 levels showed contradicting data with respect to CT [38].

In this work, we longitudinally tracked CTC numbers in M-PDAC patients enrolled in a clinical trial for niraparib, which was fast-track approved for ovarian cancer treatment [39]. Our automated CTC assay was investigated to determine the ability of CTCs to (i) track the response to therapy using a minimally invasive blood test to allow for frequent testing and (ii) prognostication to determine if patients were eligible for the therapeutic regimen.

CTC subpopulation burden, determined as the number of CTCs found in each microchip, was tracked at every PARPi treatment cycle until EOT. We found that CTC^EpCAM^ and CTC^FAPα^ burden alone were not indicative of disease status. However, the ratio of CTC^FAPα^ to CTC^EpCAM^, namely Φ, better correlated with response to therapy, as identified using CT. Gemenetzis et al. [40] used a size-based isolation of CTCs in PDAC patients and enumerated the selected subpopulations using immunophenotyping (vimentin and pan-CK expression), which indicated that the ratio of mesenchymal and epithelial phenotypes was related to recurrence. In this work, we used Φ to predict response to therapy. Unique to this work was the fact that two microchips were used, one for each subpopulation, i.e., anti-EpCAM antibodies for epithelial CTCs and FAPα for mesenchymal CTCs, which avoided the confounding results of enumeration using a continuum of phenotypes.

In 67% of patients at the EOT for whom CT indicated progressing disease, the absolute average of Φ was 5.8 (Appendix A), indicative of CTC^FAPα^ being more dominant compared with CTC^EpCAM^. We showed examples where CT correlated with Φ and CA19-9 (Figure 3C–H and Appendix A). The Kaplan–Meier plots showed the statistical advantage of ΔΦ over ΔCA19-9 in predicting tumor response; patients whose ΔΦ was >0 during treatment had a lower OS than those patients who had an instance(s) of ΔΦ < 0 (*p* = 4.95 × 10^−4^; Figure 4C). Time to EOT (i.e., PFS) was also greater in patients whose ΔΦ was <0 at least once during treatment (*p* = 1.49 × 10^−4^; Figure 4E).

Considering the necessity for tumor mutational analyses to guide therapy decisions, pancreatic tumor tissue testing using NGS does not guarantee success in identifying relevant mutations because of the complexity of the tumor microenvironment. Within a pancreatic tumor, cancer cells are a minority, while non-neoplastic cells represent 80–90% of the tumor mass [41]. Typically, a microdissection step must be performed to isolate and molecularly characterize specifically the cancer cells. Although sequencing data has identified genetic polymorphisms prevalent in PDAC [42], these studies were difficult due to elaborate sample preparation steps.

We demonstrated that CTCs can be used as a source of gDNA for sequencing to allow enrollment of patients into a clinical trial based on tumor mutational status or to evaluate potential therapy success or failure by comparing changes that occur during the treatment that may reflect chemotherapy-related evolution of different clones of cancer cells. In our studies, we observed the emergence of different mutations at the EOT. We identified multiple mutational changes in DDR genes, notably in CTC^FAPα^ and CTC^EpCAM^, with respect to mutational frequency in mechanistically related *BRCA1*, *BRCA2*, and *RAD51* genes (Figure 4, Appendix A).

*KRAS* mutations in gDNA isolated from CTCs were informative [43]. Our PCR/LDR assay confirmed that *KRAS* mutational status can be detected from CTCs. During the treatment of seven out of nine patients, we detected SNVs in *KRAS* genes from CTC^EpCAM^ and CTC^FAPα^. In the patients tested, p.G12V (c.35G>T) and p.G12A (c.35G>C) variants were detected in 77% and 11% of the patients, respectively. No p.G12R (c.34G>C) and p.G13D (c.38G>A) SNVs were detected (Figure 6D). Interestingly, gDNA did not show *KRAS* mutations in either CTC subpopulation in all cases, which can be an opportunity for combinational therapies that include anti-EGFR monoclonal antibodies, since wild-type *KRAS* cancer cells may be sensitive to these therapies [44,45].

Clinical options for PDAC primarily aim to extend patient life with the discovery of new therapeutics; thus, providing tools to accurately track response to therapy can reduce the need for continuous imaging crucial for informed treatment decisions and the discovery of new therapeutics for PDAC patients. Dual CTC selection using microfluidics allows for tracking response to therapy using a minimally invasive procedure and can enable a better understanding of disease and treatment response from the ability to determine CTC subpopulation burden and sequence CTC gDNA cargo. Our work shows that the determination of ΔΦ can be used to better assess patient outcomes when compared with CA19-9 and provide a faster assessment of tumor response to treatment compared with CT imaging. We have also shown that mutations in the DDR genetic regime can be incurred in patients undergoing PARPi treatment and are readily detectable from CTCs. Exploring exosomes for diagnosis and monitoring disease progression could be feasible in cases when the CTC burden is low. However, monitoring response to therapy for PDAC patients with metastatic disease and high CTC burden did not present challenges.

## 5. Conclusions

In this manuscript, we reported an automated CTC-based assay that consists of the selection of orthogonal subpopulations of circulating tumor cells from PDAC patients, which was used to monitor treatment response and provide genomic insight into the mutational status of the tumor during treatment. The automation was enabled using a liquid-handling robot to operate the CTC selection chips fluidically, which also improved the reproducibility of the assay. Unique to the CTC assay was the co-isolation of epithelial CTCs (EpCAM) and mesenchymal CTCs (FAPα), which better recapitulated the complex tumor microenvironment compared with using epithelial CTCs only. A special CTC selection chip was used, which consisted of a plastic chip comprised of 150 sinusoidal channels that provided high recovery, high purity, and the ability to release the cells from the capture surface. The CTC assay for monitoring response to therapy was evaluated using metastatic PDAC patients that possessed a mutation in their DNA repair genes, which was evaluated using next-generation sequencing by sourcing the DNA from CTCs. Patients who were eligible received a PARPi. Instead of using the absolute number of CTCs from either subpopulation, we used the ratio of FAPα CTCs to that of the epithelial CTCs (Φ), which performed better than CA19-9 and provided results to differentiate responders from non-responders faster than CT imaging.

## Figures and Tables

**Figure 2 cells-12-02266-f002:**
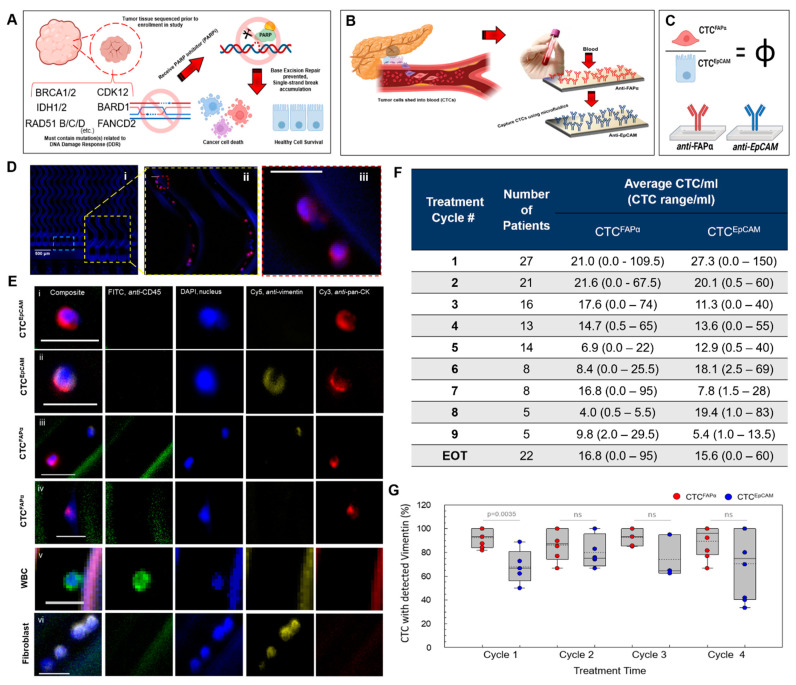
CTC isolation and enumeration. (**A**) Strategy for enrolling patients into the PARPi trial: tumor tissue gDNA is screened for DDR mutations using NGS. (**B**) Schematic showing the isolation of subpopulations of CTCs using affinity selection on a microfluidic chip. (**C**) Establishing Φ as a ratio of CTC^FAPα^ to CTC^EpCAM^. (**D**) Immunophenotyping of captured cell types: (*i*) 10× view of the 150-channel device under DAPI filter; (*ii*) 20× view of the yellow insert from (*i*), composite image of DAPI, Cy3, Cy5, and FITC; (*iii*) 40× view of the red insert from (*ii*), showing two CTCs with expression profile: DAPI (+), pan-CK (+) [Cy3], VIM (−) [Cy5], and CD45 (−) [FITC]. Scale bar in *ii* and *iii*: 25 µm. (**E**) Fluorescent images for immunostained cells *(i):* CTC^EpCAM^ isolated during cycle 1 showing CK and absence of VIM expression; (*ii*): CTC^EpCAM^ isolated during cycle 8 showing CK and VIM expression; (*iii*): Cells isolated on the anti-FAPα device, showing both a CTC (evidenced by pan-CK expression) and a cell with a profile: DAPI (+)/VIM (+)/CK (−)/CD45 (−); (*iv*). Cells isolated on the FAPα device with a CTC profile: DAPI (+)/CK (+)/VIM (−); (*v*): white blood cell showing profile: DAPI (+)/CD45 (+)/CK (−)/VIM (−); (*vi*): cells isolated on the anti-FAPα device with a profile: DAPI (+)/VIM (+)/CK (−)/CD45 (−). The cell images were collected from a chip using a fluorescent microscope. Scale bar: 20 µm. (**F**) Average and ranges of CTC^FAPα^ and CTC^EpCAM^ detected in PDAC patient blood. (**G**) Phenotyping of CTC during different treatment cycles showing the percent of CTCs with VIM expression. Paired Wilcoxon signed-rank testing was used for statistical analysis. Figure (**A**–**C**) made using BioRender, https://help.biorender.com/en/articles/3619405-how-do-i-cite-biorender (accessed on 6 July 2022).

**Figure 3 cells-12-02266-f003:**
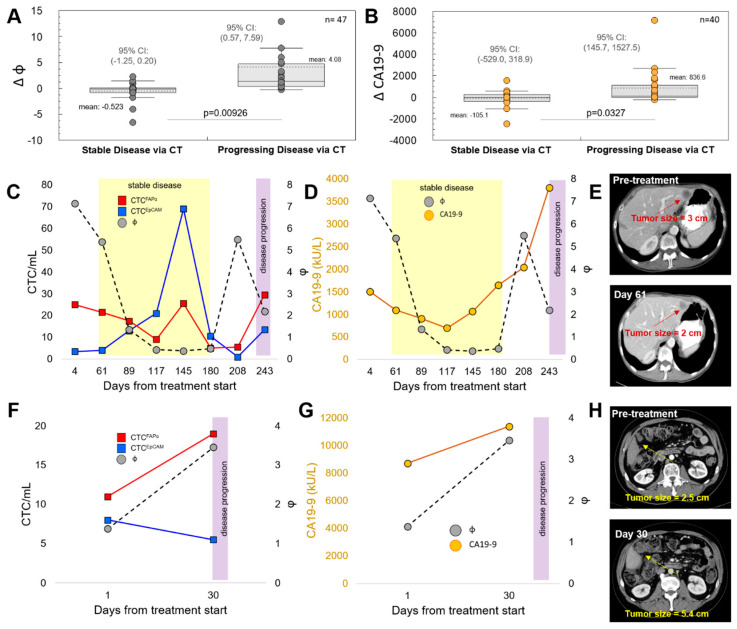
Changes in response ratio Φ versus CA19-9. (**A**) Changes in Φ and (**B**) CA19-9 levels were examined for both stable and progressing disease, as evaluated using CT. (**C**) Results from longitudinal testing of the blood collected from patient #6. Plots of CTC counts and Φ for the period that the patient underwent treatment. Yellow boxes indicate stable disease determined using CT; purple boxes indicate disease progression using CT. Stable disease was shown in both Φ and CT scans from days 61 to 180. Between days 180 and 208, the Φ ratio increased from 0.48 to 5.5. On day 243, CT showed disease progression. (**D**) Comparison between the Φ and CA19-9 trends during patient #6 longitudinal testing. (**E**) Baseline and day 61 CT images from patient #6 testing show a decrease in primary tumor size on day 61. (**F**) Results from longitudinal testing of the blood collected from patient #11. Plots of CTC counts and Φ for the period that the patient underwent treatment. The purple box indicates disease progression using CT. (**G**) Comparison between the Φ and CA19-9 trends during patient #11 longitudinal testing. (**H**) Baseline and day 30 CT images from patient #11 testing. The tumor was non-responsive to treatment, as indicated by an increase in primary tumor size after 30 days of treatment.

**Figure 4 cells-12-02266-f004:**
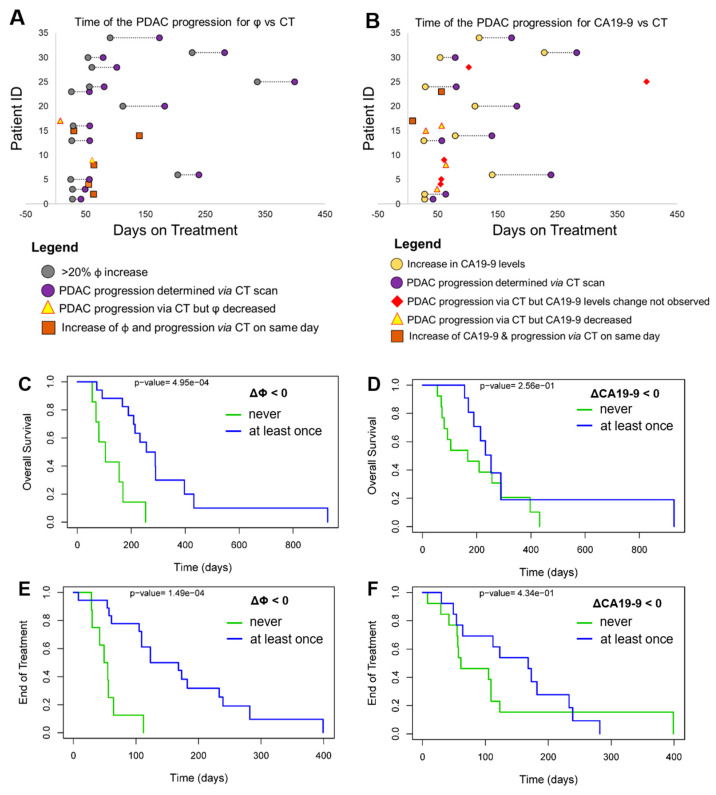
PFS and OS using CTCs vs. CA19-9. Timeline for PDAC patient testing and detection of tumor progression as observed for (**A**) ΔΦ vs. CT and (**B**) ΔCA19-9 vs. CT. Kaplan–Meier plots for overall survival (OS) for PDAC patients using (**C**) ΔΦ (n = 26) and (**D**) ΔCA19-9 (n = 26). Prediction of EOT (i.e., progression-free survival, PFS) for PDAC patients using (**E**) ΔΦ (n = 26) and (**F**) ΔCA19-9 (n = 26). Two groups of patients were evaluated: those for whom ΔΦ (n = 18) and ΔCA19-9 were never <0 (n = 13) during testing and those for whom ΔΦ (n = 8) and ΔCA19-9 (n = 13) were <0 at least once.

**Figure 5 cells-12-02266-f005:**
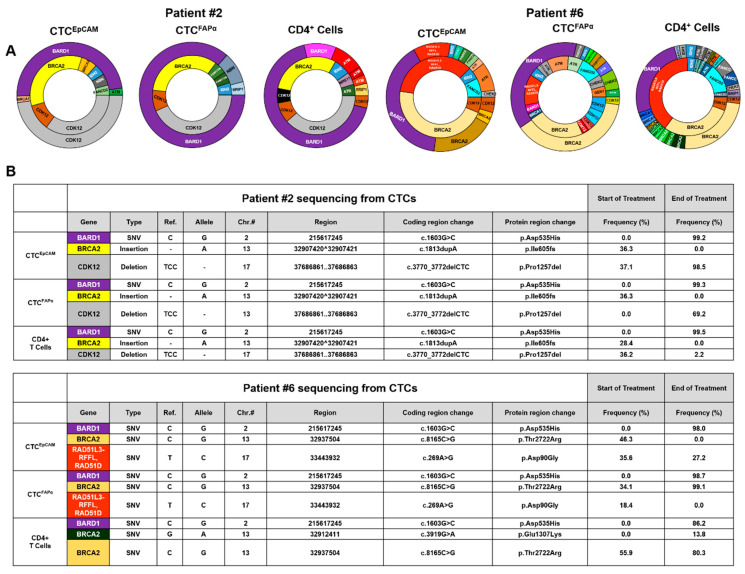
Next-generation sequencing (NGS) was performed on DNA from CTCs. (**A**) Mutation variant frequency donut plots. The inner and outer donuts represent the genomic profile of CTCs isolated at baseline and EOT, respectively. Variant profiles are shown for gDNA isolated from CTC^EpCAM^, CTC^FAPα^, and CD4^+^ cells from patients #2 and #6. Slice size is representative of variant frequency. A full list of variants is provided in Appendix A. (**B**) Table listing selected genes with mutation location, protein change, and variant frequencies at baseline and EOT for patients #2 and #6. The colors for “gene” match the colors on the donuts.

**Figure 6 cells-12-02266-f006:**
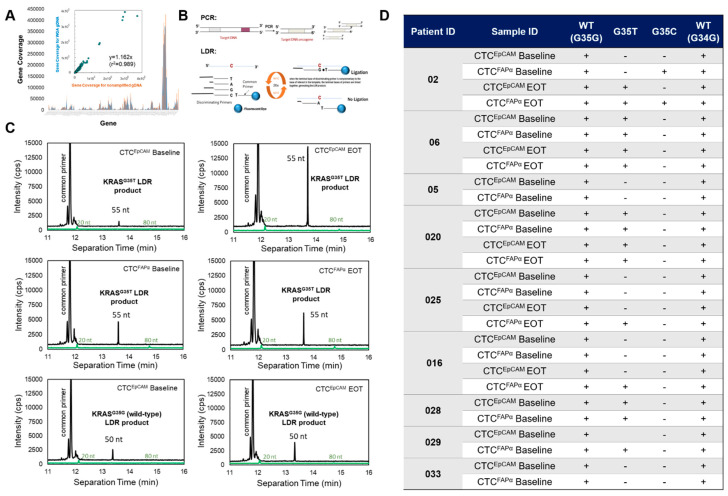
Detecting *KRAS* point mutations within CTCs. (**A**) Comparison between the NGS coverage for gDNA (10 ng, non-amplified in NGS) and WGA-gDNA (10 ng of amplified from 0.2 ng gDNA) using an Illumina AmpliSeq Focus Panel. DNA was harvested from the SKBR3 cell line. Exon sequencing from non-WGA (blue line) and WGA (orange line) shows that the gene coverage was the same (0.999). (**B**) Schematic showing the PCR/LDR assay for *KRAS* mutation screening. The PCR oncogene amplicon in LDR was linearly amplified with discriminating and common primers to detect discrete SNV; the discrete variant products had varying sizes that could be distinguished from one another using capillary gel electrophoresis. (**C**) Electropherograms were collected on a Beckman CQ CE system for the separation of wild-type (wt) and mutant-type (mt) (c.35G and c.35G>T) products following LDR. Green lines in electropherograms represent the upper and lower markers for the LDR product; black lines represent the LDR product. *KRAS* wt and mt detected in gDNA extracted from CTCs are marked in the electropherograms. The high-intensity peak at ~12 min is unligated dye-labeled common primers. *KRAS* c.35G>T (p.G12V) appeared at ~13.6 min in the electropherogram and was detected in both CTC^EpCAM^ and CTC^FAPα^ at the baseline and EOT for patient #6. Bottom-most panels show detection of wt (c.35G) control from CTC^EpCAM^. (**D**) Summary of *KRAS* mutations detected in CTCs isolated from 9 patients. (+) indicates the presence and (−) indicates the absence of LDR product.

## Data Availability

All data needed to evaluate the conclusions in this study are present in this paper and/or the Appendix A. Please reach out to authors with any questions.

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
