# Peer review of "Circulating Tumor Cell Subpopulations Predict Treatment Outcome in Pancreatic Ductal Adenocarcinoma (PDAC) Patients"

_cells, 2023, doi:10.3390/cells12182266_

Round 1
Reviewer 1 Report
This is an excellently designed study cantered around the utilization of CTCs as biomarkers for pancreatic cancer. The authors employed an innovative fluidic device coated with two sets of antibodies (EMT markers) to capture CTCs from a single tube of blood. The presented data are of exceptional quality, align well with previous findings, and overall, make coherent sense. I have a few minor inquiries and suggestions:
Have the authors attempted to assess the expression of Epcam and FAP markers on existing pancreatic cancer cell lines to validate their levels?
Could you clarify the reason behind not utilizing PAN-CK for further confirmation of CTC identification?
How would the study address potential issues if EpCAM expression was downregulated, leading to the loss of certain cells?
Regarding genomics work, have they conducted single-cell analysis as well? I presume the presented data are the result of pooled cell data upon retrieval.
Would you speculate on the potential impact of processing a larger blood volume (e.g., 20 ml) on the validity of their findings
Author Response
Reviewer 1 comments are at the end of this document.

Reviewer 2 Report
This is good work, however, a few points have to be fixed prior to acceptance.
1) This work absorbs more audience if the authors provide an appropriate graphical abstract at the end of the introduction.
2) In various research, it has been mentioned that PDAC spreads via exosomes. Cells can communicate with each other by exosomes and miRNAs. Therefore, exosomes are one level earlier than CTC. What about if the authors measured exosomes or miRNA for treatment assessment?
3) Figs 3 C, D, F, and G do not have error bars. The points should be the mean of at least three measurements with + - SD.
3) The authors did not describe the microfluidic procedure well. such as design details, fabrication procedure ...
4) How does the microfluidic chip separate CTC from WBS? Is it via physical characteristics or by affinity?
Author Response
Reviewer 2 comments are at the end of this document.

Round 2
Reviewer 2 Report
The authors answered my comments well and I would recommend publishing. The only point I would like to mention, the graphical abstract should be added at the end of the introduction part.